# Theoretical Research on Sand Penetration Grouting Based on Cylindrical Diffusion Model of Tortuous Tubes

Xuesong Wang [1], Hua Cheng [1,2,3,*], Zhishu Yao [1], Chuanxin Rong [1], Xianwen Huang [1] and Xiaoyan Liu [1]

1   School of Civil Engineering and Architecture, Anhui University of Science and Technology,
    Huainan 232001, China; xswang@aust.edu.cn (X.W.); zsyao@aust.edu.cn (Z.Y.); chxrong@aust.edu.cn (C.R.);
    huangxianwen194@163.com (X.H.); xy@aust.edu.cn (X.L.)
2   School of Resources and Environmental Engineering, Anhui University, Hefei 230601, China
3   Anhui Province Key Laboratory of Building Structure and Underground Engineering,
    Anhui Jianzhu University, Hefei 230601, China
*   Correspondence: hcheng@aust.edu.cn

**Abstract:** The penetration paths of grouts in sand layers are tortuous, and there is no reasonable penetration grouting theory at present. By employing tortuous circular tubes as the internal penetration pore channels of the sand layer, the cylindrical diffusion model is established for sand-layer penetration grouting. First, the permeability of porous media and average penetration velocity of Bingham slurry were deduced by considering tortuosity. Second, based on the penetration continuity equation, the active steady-state penetration differential equation of Bingham slurry and the pressure distribution function of slurry in the diffusion region were obtained. Finally, combined with the indoor penetration grouting test results, the attenuation law of slurry pressure and the influencing factors of diffusion radius were discussed. The results show that the cylindrical diffusion model of tortuous tubes can better characterize the diffusion process of penetration grouting in the sand layer. The stop condition for diffusion during penetration grouting is the reduction of the pressure gradient of the slurry to its starting value. The slurry pressure attenuation has obvious stages. The slurry pressure attenuation is faster in the area close to the grouting tube. At 50% of the maximum diffusion radius, the pressure attenuation is 70.01~75.41% of the total pressure attenuation within the diffusion region. Increasing the grouting pressure and permeability coefficient and reducing the slurry viscosity ratio can greatly increase the slurry diffusion radius. These results provide a theoretical basis for determining sand grouting parameters in the future.

**Keywords:** penetration grouting; tortuosity; cylindrical diffusion; Bingham slurry; diffusion radius

## 1. Introduction

Sand layer is a common, difficult to control and harmful adverse geology. Grouting method is often used in the process of subway tunnel construction through water rich sand layer, mine sand burst disaster control, near loose sand layer mining non-uniform settlement control and so on [1–3]. However, owing to many influencing factors, grouting is characterized as a complex black box problem that is invisible and intangible; thus, theoretical research on grouting lags far behind its engineering practice. Penetration grouting is the most widely used form of grouting diffusion, whose theoretical research is still incomplete. For example, existing grouting theories do not clearly define the stop criteria for slurry diffusion, and rarely consider the effect of tortuous slurry penetration paths. Therefore, it is of great significance to carry out research on the theory of penetrating grouting for perfecting the grouting theory and guiding the design of grouting parameters.

Fu et al. [4] and Zhang et al. [5] have performed extensive research on the combined use of slurries with different flow patterns and diffusion forms. Thus far, slurry flow patterns of only Newtonian, Bingham, and power-law fluids have been studied, while the diffusion forms include spherical diffusion and cylindrical diffusion [6–8]. Considering the filtration

effect of slurry diffusion, Saada et al. [9], Fan et al. [10], and You et al. [11,12] established a one-dimensional penetration grouting control equation based on the linear filtration law; they studied the influence law of filtration effect on the diffusion process of penetration grouting with laboratory tests. Maghous et al. [13] used the particle deposition–adsorption probability model to describe the deposition (filtration) behavior of slurry particles, obtained the temporal and spatial evolution law of filtration coefficient, and established the cylindrical diffusion model of penetration grouting while considering the filtration effect. By considering the time-varying viscosity of slurry, Ding et al. [14] and Li et al. [15] used the time-varying viscosity Bingham fluid constitutive equation and capillary penetration model to derive the diffusion law of penetration grouting by considering the time-space heterogeneity of slurry viscosity. Liu et al. [16,17] considered the mechanical characteristics and in-situ stress factors of the injected rock mass, analyzed the movement law of slurry rheology and crack propagation form, and established a splitting diffusion model while considering slurry rheology.

The above research results have not considered the effect of the tortuous slurry penetration path. Yu et al. [18] established a simplified model of the tortuous penetration path, and deduced an analytical solution for the fractal dimension of the tortuous penetration path in porous media. The theoretical values were in good agreement with the simulated and experimental values. Zheng et al. [19] considered the influence of pore connectivity and constructed a new capillary pressure model based on fractal penetration, which was verified by experimental data. This new model is an extension of traditional fractal theory. Du et al. [20] performed the diffusion test of cement slurry in a simulated tortuous fracture with flowing water to explore the influence of crack characteristic parameters and water flow rate on the diffusion law of cement slurry. The test results were applied to the highway damage control site. The reinforcement region before and after grouting was detected by GPR and high-density resistivity methods. The slurry completely filled the cracks in the damage region. Yun et al. [21] constructed a fractal model of the initial pressure gradient of Bingham fluid in porous media based on the fractal characteristics of pores and the capillary pressure effect, while considering that the initial pressure gradient of Bingham fluid was related to the structural parameters of porous media, yield stress of slurry, capillary pressure parameters, and fractal dimension of porous media.

In the above-mentioned research results, the one-dimensional penetration of porous media is inconsistent with the actual slurry diffusion form, and the introduced fractal theory is also used for parameter selection. Meanwhile, the stop criterion for slurry diffusion is the attenuation of pressure to zero, at which point the diffusion radius becomes maximum. To solve these problems, the penetration pore channels in sand are regarded as tortuous tubes in this study; thus, the cylindrical permeation and diffusion model of tortuous tubes for sand-layer penetration grouting was established. First, the average penetration velocity of Bingham grout was deduced by considering the tortuosity. Second, the pressure distribution function of slurry in the diffusion region was obtained according to the penetration continuity equation and boundary conditions. Finally, the influence of grout pressure attenuation law, permeability coefficient, and slurry viscosity on the diffusion radius was discussed based on the indoor penetration grouting test results. The research results are valuable for improving the grouting theory and guiding the design of grouting parameters.

## 2. Cylindrical Permeation and Diffusion Model of Tortuous Tubes

### 2.1. Conceptual Model and Basic Assumptions

Grotenhuis et al. [22,23] characterized the channel of slurry diffusion in sand by using a circular tube, and assumed that the slurry diffusion in sand pores represents the slurry flow in a circular tube. Owing to the complex pore structure of sand, the slurry diffusion in sand does not flow along a straight line, but flows forward in a tortuous way [24]. Consequently, the cylindrical permeation and diffusion model of tortuous tubes is established for sand-layer penetration grouting; that is, a certain number of circular tubes project from the axis

of the grouting tube, and the slurry penetration in sand is considered to represent the flow in circular tubes (Figure 1). In particular, the sand discussed in this research is medium sand and coarse sand with fineness modulus above 2.3 and average particle size above 0.25 mm. For fine sand or silty fine sand with small particle size, splitting grouting usually occurs because of its dense pores, which does not belong to the scope of this research.

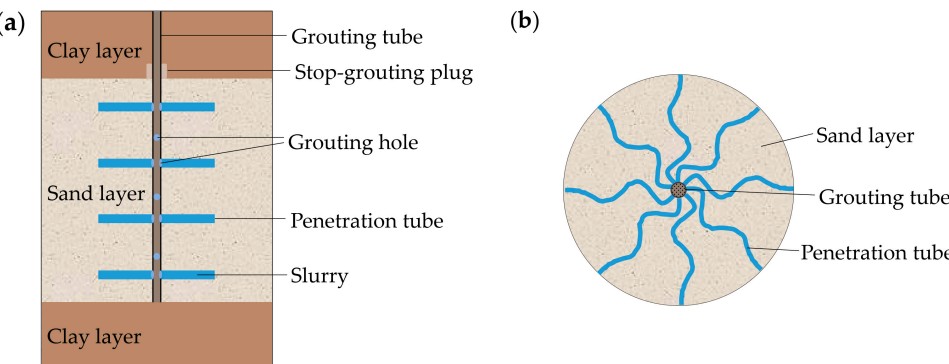

**Figure 1.** Cylindrical permeation and diffusion model of tortuous tubes: (**a**) front view; (**b**) top view.

Tortuosity is a parameter that can characterize the circuitous degree of penetration pore channels, which represents the ratio of the actual length of penetration path to the effective length of penetration path. The schematic diagram of tortuosity effect is shown in Figure 2. The definition of tortuosity is:

$$\chi = \frac{l_t}{l} \tag{1}$$

where $l_t$ is the actual length of penetration path, $l$ is the effective length of penetration path, $\chi$ is the tortuosity of the tube, and the estimated value of tortuosity according to literature is 1.41–1.58. Referring to the research results of Zhou et al. [25–27], this research assumes the tortuosity as 1.5.

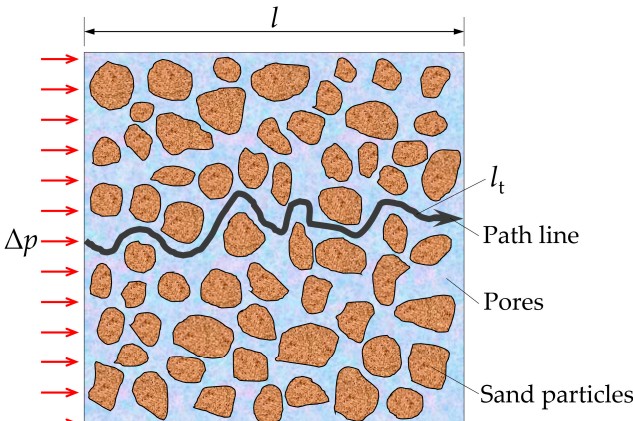

**Figure 2.** Schematic representing tortuosity.

To simplify calculations, the following assumptions have been made in this research:

1. The flow pattern of the slurry is the same as that of an incompressible Bingham fluid with time-varying viscosity, and the flow pattern remains unchanged during grouting;
2. The interaction of slurry in different tubes is not considered;
3. The inner wall of the tube satisfies the no-slip boundary condition; that is, the slurry velocity at the inner wall of the tube is 0;
4. The energy dissipation caused by the local pressure loss in the slurry flowing in the tortuous tube is ignored.

### 2.2. Flow Rate and Average Velocity of Bingham Slurry in Single Tube

Based on the above assumptions, the tortuous tube is regarded as a straight circular tube with a length of $l_t$ and a radius of $r_i$. The effective length of the penetration path is $l$, and the slurry pressure at the inlet is the same as the grouting pressure $p_0$. The coordinate system is established by assuming the tube axis to be $x$ axis, the center of the tube inlet section to be the origin, and the slurry flow direction to be the positive direction of the $x$ axis (Figure 3a).

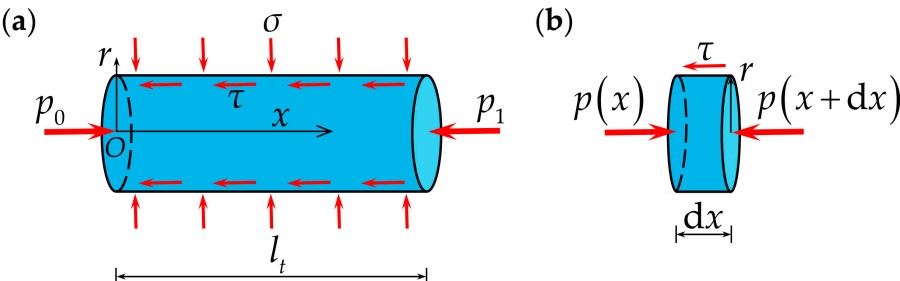

**Figure 3.** Stress analysis diagram of Bingham slurry in a tube. (**a**) The stress of the tube fluid; (**b**) fluid micro element stress.

After the slurry flows out of the grouting tube, the grouting pressure $p_0$ begins to attenuate. Assuming that the distribution function of slurry pressure along the $x$ axis is $p(x)$, the boundary conditions are

$$\begin{cases} x = r_0, \ p = p_0 \\ x = l_t, \ |\mathrm{d}p/\mathrm{d}x| = \lambda_i \end{cases} \tag{2}$$

where $r_0$ is the grouting tube radius, and $\lambda_i$ is the threshold pressure gradient of Bingham slurry flowing in a circular tube.

Take a section of cylindrical fluid micro element with radius $r$ and length $\mathrm{d}x$; its axial stress is shown in Figure 3b and obtained using the horizontal force balance condition:

$$[-p(x + \mathrm{d}x) + p(x)] \cdot \pi r^2 - \tau \cdot 2\pi r \mathrm{d}x = 0 \tag{3}$$

where $\tau$ is the shear stress between slurry layers.

Equation (3) can be rearranged as follows:

$$-[p(x + \mathrm{d}x) - p(x)] = \frac{2\tau}{r} \mathrm{d}x \tag{4}$$

If $\mathrm{d}p = p(x + \mathrm{d}x) - p(x)$, Equation (4) can be reduced into:

$$\tau = -\frac{r}{2} \frac{\mathrm{d}p}{\mathrm{d}x} \tag{5}$$

where the minus sign indicates that the pressure is decreasing; that is $\mathrm{d}p/\mathrm{d}x < 0$.

The interlaminar shear stress is proportional to the radial distance of slurry particles from the coordinate axis, and the interlaminar shear stress near the tube axis is small.

The constitutive equation of Bingham fluid is [28]:

$$\tau = \tau_0 - \eta_g \dot{\gamma} \tag{6}$$

where $\tau_0$ is the shear yield strength of slurry, $\eta_g$ is slurry viscosity, and $\dot{\gamma} = \mathrm{d}v/\mathrm{d}r < 0$ is the shear rate of slurry.

When $\tau \leq \tau_0$ and $\mathrm{d}v/\mathrm{d}r = 0$, the slurry flows in a piston-like manner as a whole. Let fluid radius be $r_p$ and velocity be $v_p$. When $\tau > \tau_0$, laminar flow occurs in the slurry.

According to Kong's research [29], the fluid radius and the fluid velocity of piston-type flow in the [0, $r_p$] range is:

$$r_p = -\frac{2\tau_0}{dp/dx} \tag{7}$$

$$v_p = \frac{1}{\eta_g}\left[-\frac{dp}{dx}\cdot\frac{r_i^2 - r_p^2}{4} - \tau_0(r_i - r_p)\right] \tag{8}$$

The fluid velocity of laminar flow in the interval [$r_p$,$r_i$] is:

$$v = \frac{1}{\eta_g}\left[-\frac{dp}{dx}\cdot\frac{r_i^2 - r^2}{4} - \tau_0(r_i - r)\right] \tag{9}$$

The velocity distribution in the circular tube is truncated paraboloid, and the single tube Bingham slurry flow rate is:

$$q_i = \frac{\pi r_i^4}{8\eta_g}\cdot\left(-\frac{dp}{dx}\right)\cdot\left[1 - \frac{4}{3}\left(\frac{2\tau_0/r_i}{-dp/dx}\right) + \frac{1}{3}\left(\frac{2\tau_0/r_i}{-dp/dx}\right)^4\right] \tag{10}$$

The average velocity on the cross section of the tube is:

$$\overline{v} = \frac{r_i^2}{8\eta_g}\cdot\left(-\frac{dp}{dx}\right)\cdot\left[1 - \frac{4}{3}\left(\frac{2\tau_0/r_i}{-dp/dx}\right) + \frac{1}{3}\left(\frac{2\tau_0/r_i}{-dp/dx}\right)^4\right] \tag{11}$$

At $\overline{v} = 0$, the threshold pressure gradient $\lambda_i$ of Bingham slurry flow in a circular tube is:

$$\lambda_i = \left|\frac{dp}{dx}\right| = \frac{2\tau_0}{r_i} \tag{12}$$

By substituting Equation (12) into Equation (11), we obtain the following expression:

$$\overline{v} = \frac{r_i^2}{8\eta_g}\cdot\left(-\frac{dp}{dx}\right)\cdot\left[1 - \frac{4}{3}\left(\frac{\lambda_i}{-dp/dx}\right) + \frac{1}{3}\left(\frac{\lambda_i}{-dp/dx}\right)^4\right] \tag{13}$$

*2.3. Permeability of Porous Media Considering Tortuosity*

The process of theoretical derivation renders it difficult to determine the tube number $N_i$ and tube radius $r_i$, which need to be transformed into sand permeability $k$ since it is easy to measure. The following derivation uses parameters $N_i$ and $r_i$ to characterize permeability $k$.

Permeability $k$ is not dependent on slurry characteristics; however, it is affected by porosity, capillary diameter, and tortuosity of porous media. The existing theory only uses porosity and capillary diameter to characterize the permeability of porous media [29]; however, it does not consider tortuosity, which should not be ignored. It is worth noting that because permeability does not affect fluid characteristics, a laminar Newtonian fluid has been used in the permeability process to simplify the calculation; this is not in contradiction with the Bingham fluid discussed in this study.

To ensure that the model closely resembles the actual porous media, the tube radius is non-uniform. Based on Hagen–Poiseuille equation [30], the flow rate through all tubes per unit time while considering tortuosity is:

$$q = -\frac{\sum\limits_{i=1}^{N} N_i\pi r_i^4\Delta p}{8\eta_g\chi l} \tag{14}$$

where $q$ is the flow through the pore section, $N_i$ is the number of tubes with radius $r_i$ ($i = 1$, 2, ... $N$), and $\Delta p$ is the pressure difference between the two ends of the tube.

Under the laminar flow state, the slurry penetration in porous media meets Darcy's law, and the flow rate through the pore section per unit time is:

$$q = -\frac{kA\Delta p}{\eta_g l} \tag{15}$$

where $k$ is the permeability of the porous media and $A$ is the penetration cross-sectional area.

Through theoretical derivation, Li et al. [31] proved that tube flow and penetration flow are completely equivalent; the formulas of the two mathematical models are also completely equivalent, but they are expressed by different parameters. By combining Equations (14) and (15), the permeability of porous media while considering tortuosity is obtained:

$$k = \frac{\sum\limits_{i=1}^{N} N_i \pi r_i^4}{8\chi A} \tag{16}$$

The effective porosity of the tube penetration model is:

$$\varphi' = \frac{\pi\chi \sum\limits_{i=1}^{N} N_i r_i^2}{A} \tag{17}$$

where $\varphi'$ is the effective porosity of porous media; that is, the ratio of the pore volume filled by slurry to the total volume of porous media. The theoretical calculation formula associated with permeability of porous media is calculated while considering tortuosity:

$$k = \frac{\varphi'}{8\chi^2} \cdot \frac{\sum\limits_{i=1}^{N} N_i r_i^4}{\sum\limits_{i=1}^{N} N_i r_i^2} \tag{18}$$

### 2.4. Slurry Pressure Distribution of Cylindrical Penetration and Diffusion

For a one-dimensional penetration of slurry in sandy soil, the total flow rate in all tubes can be obtained from Equation (10):

$$q = \frac{\pi}{8\eta_g} \cdot \left(-\frac{\mathrm{d}p}{\mathrm{d}x}\right) \cdot \left[\sum_{i=1}^{N} N_i r_i^4 - \frac{4}{3}\left(\frac{2\tau_0}{-\mathrm{d}p/\mathrm{d}x}\right)\sum_{i=1}^{N} N_i r_i^3 + \frac{1}{3}\left(\frac{2\tau_0}{-\mathrm{d}p/\mathrm{d}x}\right)^4 \sum_{i=1}^{N} N_i\right] \tag{19}$$

To simplify Equation (19) based on the statistical principle and the method adopted by Kong [29], the average radius $\bar{r}$ of tubes is determined:

$$\bar{r} = \frac{\sum\limits_{i=1}^{N} N_i r_i^4}{\sum\limits_{i=1}^{N} N_i r_i^3} \approx \left(\frac{\sum\limits_{i=1}^{N} N_i r_i^4}{\sum\limits_{i=1}^{N} N_i}\right)^{\frac{1}{4}} \tag{20}$$

Then,

$$q = \frac{\pi}{8\eta_g} \cdot \sum_{i=1}^{N} N_i r_i^4 \cdot \left(-\frac{\mathrm{d}p}{\mathrm{d}x}\right) \cdot \left[1 - \frac{4}{3}\left(\frac{2\tau_0/\bar{r}}{-\mathrm{d}p/\mathrm{d}x}\right) + \frac{1}{3}\left(\frac{2\tau_0/\bar{r}}{-\mathrm{d}p/\mathrm{d}x}\right)^4\right] \tag{21}$$

From Equations (17) and (21), the penetration velocity can be obtained:

$$v = \frac{q}{A} = \frac{\varphi'}{8\chi\eta_g} \cdot \frac{\sum\limits_{i=1}^{N} N_i r_i^4}{\sum\limits_{i=1}^{N} N_i r_i^2} \cdot \left(-\frac{dp}{dx}\right) \cdot \left[1 - \frac{4}{3}\left(\frac{2\tau_0/\bar{r}}{-dp/dx}\right) + \frac{1}{3}\left(\frac{2\tau_0/\bar{r}}{-dp/dx}\right)^4\right] \tag{22}$$

If $v = 0$, the average threshold pressure gradient $\overline{\lambda}$ of Bingham fluid penetration is:

$$\overline{\lambda} = \left|\frac{dp}{dx}\right| = \frac{2\tau_0}{\bar{r}} \tag{23}$$

By substituting Equations (18) and (23) into Equation (22), the penetration velocity can be transformed into:

$$v = \frac{\chi k}{\eta_g} \cdot \left(-\frac{dp}{dx}\right) \cdot \left[1 - \frac{4}{3}\left(\frac{\overline{\lambda}}{-dp/dx}\right) + \frac{1}{3}\left(\frac{\overline{\lambda}}{-dp/dx}\right)^4\right] \tag{24}$$

The relationship [32] between the fluid permeability coefficient $K$ and the permeability $k$ of porous media is:

$$K = k\frac{\gamma}{\eta} \tag{25}$$

where $K$ is the fluid permeability coefficient, $k$ is the permeability of porous media, $\gamma$ is fluid gravity, and $\eta$ is fluid viscosity.

The viscosity ratio of slurry to water is:

$$\beta = \frac{\eta_g}{\eta_w} \tag{26}$$

Then,

$$K_g = \frac{\gamma_g}{\beta\gamma_w} K_w \tag{27}$$

where $K_g$ is the permeability coefficient of slurry and $K_w$ is the permeability coefficient of water.

By substituting Equations (25) to (27) into Equation (24), the penetration velocity can be further reduced to:

$$v = -\frac{\chi K_w}{\beta\gamma_w} \cdot \frac{dp}{dx} \cdot \left[1 - \frac{4}{3}\left(\frac{\overline{\lambda}}{-dp/dx}\right) + \frac{1}{3}\left(\frac{\overline{\lambda}}{-dp/dx}\right)^4\right] \tag{28}$$

Referring to the theory of Kong et al. [29], Equation (28) can be extended to obtain the three-dimensional penetration velocity:

$$v = -\frac{\chi K_w}{\beta\gamma_w} \cdot \nabla p \cdot \left[\left(1 - \frac{\overline{\lambda}}{|\nabla p|}\right) - \frac{1}{3} \cdot \frac{\overline{\lambda}}{|\nabla p|}\left(1 - \frac{\overline{\lambda}^3}{|\nabla p|^3}\right)\right] \tag{29}$$

If

$$\frac{1}{|\nabla p|} = \Theta(x, y, z, t), \tag{30}$$

then

$$v = -\frac{\chi K_w}{\beta\gamma_w} \cdot \nabla p \cdot \left[(1 - \overline{\lambda}\Theta) - \frac{1}{3}\overline{\lambda}\Theta\left(1 - \overline{\lambda}^3\Theta^3\right)\right] \tag{31}$$

The second term in the square brackets of Equation (31) can be considered negligible relative to the first term [29]:

$$v = -\frac{\chi K_\text{w}}{\beta \gamma_\text{w}} \cdot \nabla p \cdot \left(1 - \bar{\lambda}\Theta\right) \tag{32}$$

Then, the divergence of penetration velocity is:

$$\nabla \cdot v = -\frac{\chi K_\text{w}}{\beta \gamma_\text{w}} \left[\left(1 - \bar{\lambda}\Theta\right) \nabla^2 p - \bar{\lambda}\nabla\Theta \cdot \nabla p\right] \tag{33}$$

For the active steady-state penetration of incompressible fluid, according to the law of mass conservation, the penetration continuity equation is [33]:

$$-\frac{\chi K_\text{w}}{\beta \gamma_\text{w}} \left[\left(1 - \bar{\lambda}\Theta\right) \nabla^2 p - \bar{\lambda}\nabla\Theta \cdot \nabla p\right] + q = 0 \tag{34}$$

The active steady-state penetration differential equation of Bingham fluid is:

$$\left(1 - \bar{\lambda}\Theta\right) \nabla^2 p - \bar{\lambda}\nabla\Theta \cdot \nabla p = \frac{q\beta\gamma_\text{w}}{\chi K_\text{w}} \tag{35}$$

For cylindrical diffusion of Bingham slurry,

$$\begin{cases} \nabla p = \frac{\partial p}{\partial \rho} < 0 \\ \nabla^2 p = \frac{1}{\rho}\frac{\partial p}{\partial \rho} + \frac{\partial^2 p}{\partial \rho^2} \\ \Theta = -1/\frac{\partial p}{\partial \rho} \\ \nabla\Theta = \frac{\partial^2 p}{\partial \rho^2} / \left(\frac{\partial p}{\partial \rho}\right)^2 \end{cases}. \tag{36}$$

where $\rho$ is the radial distance from the grouting tube axis.

Then, Equation (35) turns into:

$$\frac{\partial^2 p}{\partial \rho^2} + \frac{1}{\rho}\frac{\partial p}{\partial \rho} + \frac{\bar{\lambda}}{\rho} = \frac{q\beta\gamma_\text{w}}{\chi K_\text{w}} \tag{37}$$

The general solution of the differential equation is:

$$p(\rho) = C_1 \ln \rho - \frac{q\beta\gamma_\text{w}}{4\chi K_\text{w}}\rho^2 - \bar{\lambda}\rho + C_2 \tag{38}$$

where $C_1$ and $C_2$ are arbitrary constants.

The boundary conditions are:

$$\begin{cases} \rho = r_0 , \ p = p_0 \\ \rho = R , \ |\mathrm{d}p/\mathrm{d}\rho| = \bar{\lambda} \end{cases} \tag{39}$$

where $R$ is the diffusion radius of slurry.

By substituting Equation (39) into Equation (38), the constant terms are equal to:

$$\begin{cases} C_1 = -\frac{q\beta\gamma_\text{w}R^2}{2\chi K_\text{w}} \\ C_2 = p_0 + \frac{q\beta\gamma_\text{w}R^2}{2\chi K_\text{w}}\ln r_0 - \frac{q\beta\gamma_\text{w}}{4\chi K_\text{w}}r_0^2 + \bar{\lambda}r_0 \end{cases} \tag{40}$$

Then, the radial slurry pressure distribution in the diffusion region is:

$$p(\rho) = p_0 - \frac{q\beta\gamma_\text{w}R^2}{2\chi K_\text{w}}\ln\frac{\rho}{r_0} + \frac{q\beta\gamma_\text{w}}{4\chi K_\text{w}}\left(\rho^2 - r_0^2\right) - \bar{\lambda}(\rho - r_0) \tag{41}$$

## 3. Results and Discussion

### 3.1. Attenuation Law of Slurry Pressure

The indoor infiltration grouting test conducted by Li [34] was used to discuss the cylindrical permeation and diffusion model of Bingham fluid in a tortuous tube that was proposed in this research. Li conducted 10 sand permeation grouting tests using a self-made test device that simulates grouting. The test device mainly includes a grouting system, model test box, and monitoring system. The slurry is prepared by mixing a special sand-fixing agent used for the subway tunnel with water according to the designed water-binder ratio, and the injected material is prepared using sand in the grouting layer of Shenyang Metro. The grouting parameters and test results are shown in Table 1.

**Table 1.** Grouting parameters and test results.

| Test No. | Grouting Pressure (kPa) | Permeability Coefficient ($10^{-4}$ m·s$^{-1}$) | Water-Binder Ratio | Average Viscosity (mPa·s) | Diffusion Radius (cm) |
|---|---|---|---|---|---|
| ZJ1 | 30 | 3.17 | 3.8 | 73.10 | 8.1 |
| ZJ2 | 40 | 2.46 | 3.5 | 71.01 | 7.5 |
| ZJ3 | 50 | 1.88 | 3.2 | 88.41 | 7.0 |
| ZJ4 | 60 | 1.31 | 4.0 | 74.37 | 9.6 |
| ZJ5 | 70 | 0.76 | 3.7 | 71.71 | 6.9 |
| ZJ6 | 80 | 3.54 | 3.4 | 94.39 | 13.5 |
| ZJ7 | 90 | 2.79 | 3.1 | 106.33 | 12.9 |
| ZJ8 | 100 | 2.20 | 3.9 | 80.00 | 13.7 |
| ZJ9 | 110 | 1.58 | 3.6 | 80.81 | 12.1 |
| ZJ10 | 120 | 1.06 | 3.3 | 85.29 | 8.2 |

The slurry flow pattern is the same as that of a Bingham fluid. Through trial calculation, the time-varying law of slurry viscosity conforms to the following exponential function:

$$\eta = \eta_0 e^{at} \tag{42}$$

where $\eta$ is the slurry viscosity, $\eta_0$ is the initial viscosity of slurry, and a is the time-varying coefficient of viscosity.

The measured viscosity values of the three kinds of slurry with water–binder ratios of 3.0, 3.5 and 4.0 are re-fitted, and the fitting results are shown in Figure 4. The certainty coefficients of goodness of fit ($R^2$) are 0.97431, 0.93291, and 0.97455, while exhibiting a high fitting accuracy. It is worth noting that the slurry preparation involves the selection of a special sand consolidation agent for the subway tunnel. Its viscosity is still high when the water–binder ratio is above 3.0, and its flow pattern is the same as that of a Bingham fluid, which is obviously different from that of cement slurry when the water-binder ratio exceeds 3.0. Then, using the approximate method of Shamu et al. [35], the initial viscosities of slurries with different water–binder ratios in each grouting test are obtained via linear interpolation along with the viscosity at the end of grouting. The average of the two viscosities is taken as the average viscosity of slurry. The calculation results are shown in Table 1.

The viscosity $\eta_w$ of water is 1.0 mPa·s, gravity of water is 10.0 kN/m$^3$, radius of grouting tube is 1.0 cm, and grouting flow $q$ is 0.06 L/min. By substituting relevant parameters into Equation (41), 10 groups of slurry pressure attenuation curves in the diffusion region of the indoor penetration grouting test are obtained (Figure 5).

The absolute value of the slope of $p - \rho$ curve is represented by the slurry pressure attenuation gradient. Figure 5 shows that the pressure attenuation of slurry occurs in stages: the pressure attenuation of the slurry near the grouting tube occurs faster; that is, the pressure gradient is larger. The slurry pressure in the region far away from the grouting tube tends to decrease linearly; that is, the pressure gradient is approximately constant and attenuates slowly, which is consistent with the pressure decreasing trend observed by relevant scholars [36,37]. Through calculation, when the grouting pressure is

30~70 kPa, the slurry pressure attenuation at 50% of the diffusion radius is 70.37%, 70.01%, 70.08%, 73.00%, and 70.49% of the total pressure attenuation within the diffusion region respectively. When the grouting pressure is 80~120 kPa, the slurry pressure attenuation at 50% of the diffusion radius is 75.02%, 75.00%, 75.41%, 74.73%, and 71.90% of the total pressure attenuation within the diffusion region, respectively. That is, when the slurry diffuses to 50% of the maximum diffusion radius, the pressure attenuation is 70.01~75.41% of the total pressure attenuation within the diffusion region. In the remaining 50% diffusion region, the pressure decreases the remaining 24.59~29.99% approximately linearly.

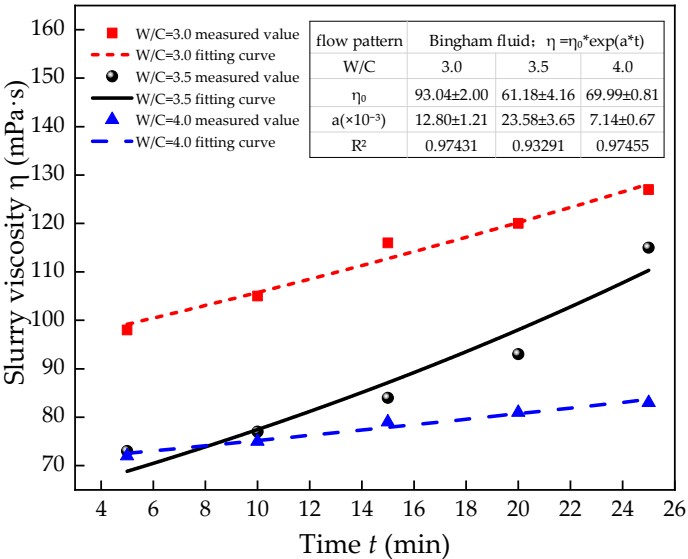

| flow pattern | Bingham fluid: $\eta = \eta_0 * \exp(a*t)$ | | |
| --- | --- | --- | --- |
| W/C | 3.0 | 3.5 | 4.0 |
| $\eta_0$ | 93.04±2.00 | 61.18±4.16 | 69.99±0.81 |
| $a(\times 10^{-3})$ | 12.80±1.21 | 23.58±3.65 | 7.14±0.67 |
| $R^2$ | 0.97431 | 0.93291 | 0.97455 |

**Figure 4.** Time-varying curve of slurry viscosity fitted by Equation (42).

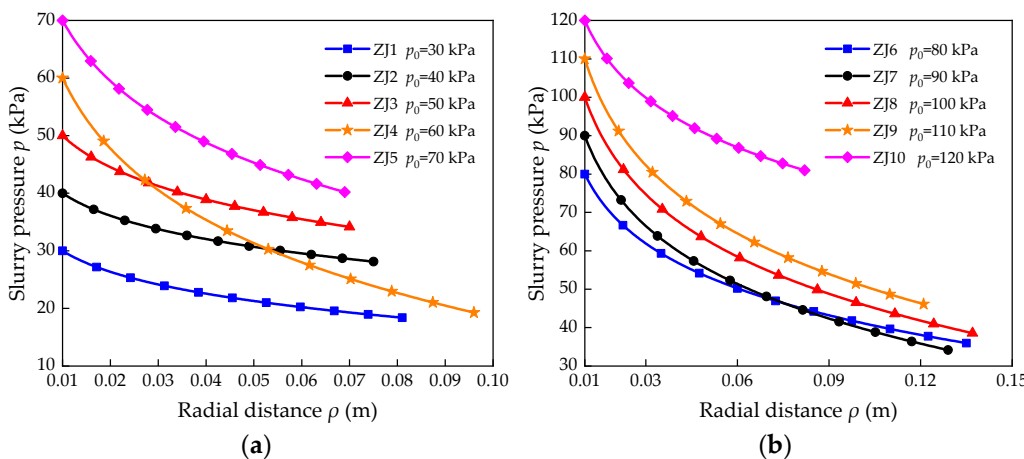

**Figure 5.** Pressure attenuation curve in the permeation and diffusion region, which was drawn by Equation (41): (**a**) ZJ1–ZJ5; (**b**) ZJ6–ZJ10.

The reasons for these results are as follows: the slurry velocity is faster in the region near the grouting tube, whereas it is slower in the region far away from the grouting tube. According to Equation (32), the absolute value of the slurry pressure gradient is positively correlated with penetration velocity. Therefore, along the direction of slurry diffusion, the pressure gradient attenuates continuously; that is, the pressure attenuates more and more slowly. When the slurry pressure gradient attenuates to the threshold pressure gradient $\overline{\lambda}$, the slurry stops spreading and the diffusion distance $R$ at this time is the diffusion radius of penetration grouting. Some scholars [38,39] believe that when the grout pressure attenuates to zero, or the hydrostatic pressure at this point becomes zero, or the principal stress of

the soil layer is minimum, the slurry stops spreading and reaches its maximum diffusion radius. In other words, it is considered that the magnitude of slurry pressure or pressure difference is the control condition of whether slurry can continue to penetrate and diffuse; however, there is no theoretical basis for this hypothesis. Equation (41) and Figure 5 reveal that the slurry pressure at the diffusion radius is not attenuated to zero, or the hydrostatic pressure or the minimum principal stress. Here, the stop criterion for slurry diffusion is whether the slurry pressure gradient attenuates to the threshold pressure gradient; that is, the slurry pressure stops diffusion before it attenuates to zero. In previous studies, unreasonable stop criteria for slurry diffusion have led to extremely high theoretical values of the grouting diffusion radii [40–42]. According to Equation (23), the magnitude of the threshold pressure gradient depends on the shear yield strength of slurry and the average radius of penetration tube in the injected formation. The shear yield strength of slurry hardly changes with time during grouting [43], and it is easy to measure. However, it is difficult to measure the average radius of the penetration tube in the injected layer due to the heterogeneity of the injected layer and the penetration effect of slurry in the diffusion process. For different injected soils, the measurement methods and specific values need to be further researched.

### 3.2. Analysis of Influencing Factors of Diffusion Radius

#### 3.2.1. Fitting between Diffusion Radius and Grouting Parameters

By fitting the data in Table 1, the mathematical model of the relationship between diffusion radius and grouting parameters is set as:

$$R = b p_0^c K_w^d \beta^f \tag{43}$$

where $R$ is in cm; $p_0$ is in kPa; $K_w$ is $10^{-4}$ m/s; and b, c, d, and f are fitting coefficients.

The fitting result is:

$$R = 5.203 p_0^{0.622} K_w^{0.533} \beta^{-0.534} \tag{44}$$

The $R^2$ value is 0.96 at a high fitting accuracy, which can be better used to describe the relationship between slurry diffusion radius and grouting parameters.

#### 3.2.2. Influence of Sand Permeability Coefficient on Diffusion Radius

The influence of sand permeability coefficient on grouting diffusion radius was analyzed using the sand permeability coefficients in Table 1. Figure 6 shows the $R$-$p_0$ relationship curves under five different parameters: the sand permeability coefficients are $0.75 \times 10^{-4}$ m/s, $1.50 \times 10^{-4}$ m/s, $2.25 \times 10^{-4}$ m/s, $3.00 \times 10^{-4}$ m/s, and $3.75 \times 10^{-4}$ m/s, and the slurry viscosity ratio is 90. It can be seen from Figure 6 that when other parameters are the same, the slurry diffusion radius increases nonlinearly with the increase of grouting pressure. When the grouting pressure is low, the diffusion radius grows rapidly. When the grouting pressure rises to a certain value, the growth rate of diffusion radius slows down. When other parameters are the same, the slurry diffusion radius increases nonlinearly with the increase of permeability coefficient. When the permeability coefficient is low, the diffusion radius increases rapidly. When the permeability coefficient increases to a certain value, the growth rate of diffusion radius slows down. Increase the grouting pressure and permeability coefficient together, and the diffusion radius will be greatly increased. When the permeability coefficient is changed from $0.75 \times 10^{-4}$ m/s to $1.50 \times 10^{-4}$ m/s, and the grouting pressure is changed from 100 kPa to 150 kPa, the grouting diffusion radius can be increased by 86.07%. An appropriate increase in grouting pressure at the project site can increase the region of grouting reinforcement; however, it is also necessary to prevent the ground uplift or destruction of stratum structures caused by splitting grouting due to too high grouting pressures. With a linear increase in the sand permeability coefficient, the diffusion radius of slurry increases nonlinearly under the same grouting pressure, and the increment decreases step by step. A water injection test is often carried out before formal grouting is performed on site. Thus, it is imperative to obtain the relevant param-

eters of sand layers, which provide a basis for grouting parameter design. Furthermore, it is important to increase the stratum permeability coefficient and improve the sensitivity of diffusion radius to grouting pressure. Therefore, the water injection test can not only increase the formation permeability coefficient, but also increase the grout diffusion radius and expand the grouting reinforcement range.

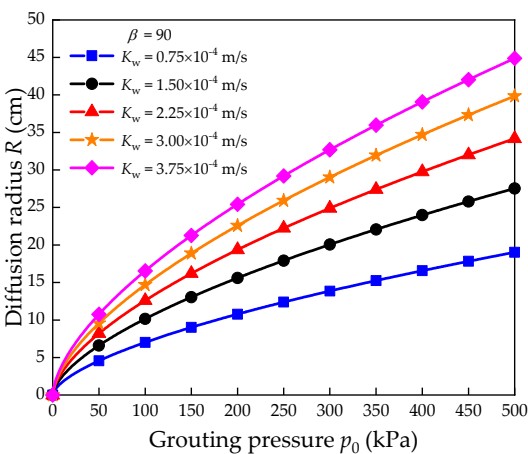

**Figure 6.** Influence of sand permeability coefficient on diffusion radius, which was drawn by Equation (44).

### 3.2.3. Influence of Slurry Viscosity on Diffusion Radius

To analyze the influence of slurry viscosity on grouting diffusion radius, the average slurry viscosity shown in Table 1 is used. Figure 7 shows the $R$-$p_0$ relation curves under five different parameters: the slurry viscosity ratio is 70, 80, 90, 100 and 110, and the sand permeability coefficient is $2.25 \times 10^{-4}$ m/s. It can be seen from Figure 7 that with the linear decrease of slurry viscosity ratio, the increase of slurry diffusion radius increases slightly under the same grouting pressure. While reducing the slurry viscosity ratio and increasing the grouting pressure, the grouting diffusion radius can be greatly improved. When the slurry viscosity ratio is changed from 110 to 70, and the grouting pressure is changed from 100 kPa to 150 kPa, the grouting diffusion radius can be increased by 63.70%. Therefore, on the premise of satisfying economy and safety, reducing the slurry viscosity and increasing the grouting pressure can greatly improve the diffusion radius.

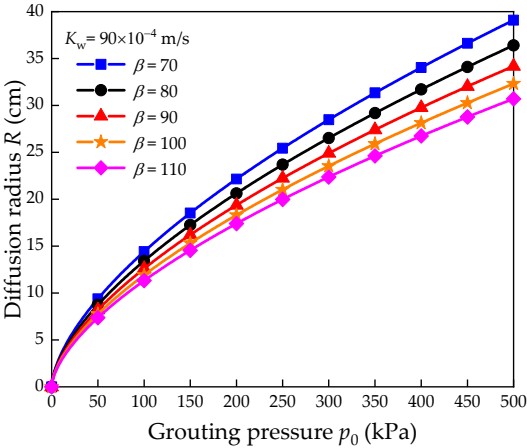

**Figure 7.** Influence of slurry viscosity on diffusion radius, which was drawn by Equation (44).

### 4. Conclusions

To explore the theory of sand penetration grouting considering tortuous effect, this research regards the penetration pore channels in sand as tortuous circular tubes, establishes the cylindrical diffusion model of tortuous tubes for sand penetration grouting, and obtains

the pressure distribution function of slurry in the diffusion region. Combined with the results of indoor penetration grouting test, the attenuation law of slurry pressure in the diffusion region and the effects of permeability coefficient and slurry viscosity on grouting diffusion radius are discussed. The following conclusions are obtained:

(1) Based on the diffusion model of the Bingham slurry in a single tube, the average one-dimensional penetration velocity of the Bingham slurry in sand was obtained and extended to three-dimensional penetration. Based on the penetration continuity equation, the active steady-state penetration differential equation of Bingham fluid and the pressure distribution function of slurry in the diffusion region were obtained.

(2) The pressure attenuation of slurry shows obvious stages. The slurry pressure attenuation is fast in the region close to the grouting tube, and the slurry pressure in the region far away from the grouting tube tends to decrease linearly. When the slurry diffuses to 50% of the maximum diffusion radius, the pressure attenuation is 70.01~75.41% of the total pressure attenuation within the diffusion region. In the remaining 50% diffusion region, the pressure decreases the remaining 24.59~29.99% approximately linearly.

(3) The stop criterion of slurry diffusion is the attenuation of the slurry pressure gradient to its threshold value. The slurry diffusion radius increases nonlinearly with the increase of grouting pressure and permeability coefficient. When the permeability coefficient is changed from $0.75 \times 10^{-4}$ m/s to $1.50 \times 10^{-4}$ m/s, and the grouting pressure is changed from 100 kPa to 150 kPa, the grouting diffusion radius can be increased by 86.07%. While reducing the slurry viscosity ratio and increasing the grouting pressure, the grouting diffusion radius can be greatly improved. When the slurry viscosity ratio is changed from 110 to 70, and the grouting pressure is changed from 100 kPa to 150 kPa, the grouting diffusion radius can be increased by 63.70%.

**Author Contributions:** Conceptualization, X.W. and H.C.; methodology, X.W. and C.R.; software, X.H.; validation, X.W. and X.H.; formal analysis, X.L.; investigation, X.L.; resources, H.C. and Z.Y.; data curation, X.W., and X.H.; writing—original draft preparation, X.W.; writing—review and editing, X.W.; visualization, X.W.; supervision, Z.Y.; project administration, C.R.; funding acquisition, H.C. All authors have read and agreed to the published version of the manuscript.

**Funding:** This research was funded by: National Natural Sciences Foundation of China: 51874005; National Natural Sciences Foundation of China: 51674006; National Natural Sciences Foundation of China: 51878005.

**Institutional Review Board Statement:** Not applicable.

**Informed Consent Statement:** Not applicable.

**Data Availability Statement:** Data used in this research are duly available from the first authors on reasonable request.

**Acknowledgments:** The authors are grateful to the Editor and anonymous reviewers who played a significant role in shaping and improving the manuscript.

**Conflicts of Interest:** The authors declare no conflict of interest.

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
