# Peer review of "Theoretical Research on Sand Penetration Grouting Based on Cylindrical Diffusion Model of Tortuous Tubes"

_water, doi:10.3390/w14071028_

Round 1

Reviewer 1 Report

This research is interesting. There are some concerns that need to be considered.

  1. Abstract; “Under the same grouting pressure, …”, the result presented in this statement is general and well-known. Instead give values to show how much is the penetration. For example, the results presented in line 346. 
  2. Give an introduction and the theory behind the “tortuous penetration path” that was considered in this study. This is important to be presented in the introduction as it is the research gap claimed by the authors.
  3. Line 69; “test results were successful …”, describe more about being successful on the site. This is also important because when study is just in theory it is important to know how it will work in practice and how much is error in prediction/performance.
  4. Grouting of “sand layers” is considered in this research. Please give examples where we need to do grouting for sand layers in practice? It can be presented in the introduction to justify the importance of this research.   
  5. Section 2.1; what is the definition of “sand” that was assumed in this investigation? Can you specify based on a standard or specific particle sizes?
  6. In the figures caption, specify that the curves were plotted based on which equation, give the equation number.
  7. For X- and Y- axis, please mention the complete name with the unit in the parenthesis.
  8. Give a short introduction in section 4 before presenting the specific conclusions.
  9. Conclusion 3; there is no discussion about nonlinear correlation, then how can make a conclusion? 
  10. Conclusions should include some important data based on the results extracted from the models.

Reviewer 2 Report

The paper has presented a numerical research on grouting penetration, using cylindrical diffusion model of tortuous tubes. My biggest concern is not about the research outcomes, but its scientific contribution. The penetration of Bingham slurry has been studied well with various theories. Author should clearly state their new theoretical contribution in abstract and conclusion.

Here are some minor comments:

  • References are incorrect. The most critical citation is [34]. It should belong to Li, but it belongs to Shamu in the list.
  • As the research employs some data from Li, they should confirm that there is not interest conflict or any problem with copyright.
  • The result fitting is fine, but the influence has not been validated.
  • Conclusion 2 and 3 must be quantified. Readers cannot apply these conclusion to their work.

Round 2

Reviewer 1 Report

Thanks for considering the comments and improving the paper. However, in the case of my comment #7, the names of X-axis and Y-axis of Figures 4 ~ 7 are still parametric and not full names. Then in this way of presentation a reader will be forced to look for definitions in the paper. Figures should be independent.

Author Response

Thank you for your valuable comments. I’m very sorry that the problems mentioned in comment 7 were not well solved in the last round of response. According to the suggestion of reviewer, the names of X-axis and Y-axis in Figures 4 ~ 7 have been changed to full names in the newly submitted manuscript. We tried our best to improve the manuscript and made some changes. We appreciate for Editors/Reviewers’ warm work earnestly, and hope that the correction will meet with approval. If anything incorrect appears, please contact us in time, we will correct it. Special thanks to you for your good comments.